# Intermediate Judgments and Trust in Artificial Intelligence-Supported Decision-Making

**DOI:** 10.3390/e26060500

**Published:** 2024-06-08

**Authors:** Scott Humr, Mustafa Canan

**Affiliations:** Department of Information Sciences, Naval Postgraduate School, Monterey, CA 93943, USA; mustafa.canan@nps.edu

**Keywords:** artificial intelligence, decision-making, trust, quantum decision theory, quantum open systems modeling

## Abstract

Human decision-making is increasingly supported by artificial intelligence (AI) systems. From medical imaging analysis to self-driving vehicles, AI systems are becoming organically embedded in a host of different technologies. However, incorporating such advice into decision-making entails a human rationalization of AI outputs for supporting beneficial outcomes. Recent research suggests intermediate judgments in the first stage of a decision process can interfere with decisions in subsequent stages. For this reason, we extend this research to AI-supported decision-making to investigate how intermediate judgments on AI-provided advice may influence subsequent decisions. In an online experiment (N = 192), we found a consistent bolstering effect in trust for those who made intermediate judgments and over those who did not. Furthermore, violations of total probability were observed at all timing intervals throughout the study. We further analyzed the results by demonstrating how quantum probability theory can model these types of behaviors in human–AI decision-making and ameliorate the understanding of the interaction dynamics at the confluence of human factors and information features.

## 1. Introduction

Humans are increasingly using artificial intelligence (AI) to support decision-making in a variety of ways. From medical imaging to self-driving vehicles, AI is proactively operating on information in ways to support a variety of decision processes [1]. AI outputs, however, are still not well understood [2], and may influence decision outcomes in deleterious ways. Decisions supported by AI have resulted in the killing of innocent individuals [3] in some instances. For these reasons, understanding how AI technologies can inadvertently affect human decision-making processes is necessary for improving decision outcomes.

Understanding how AI outputs influence the human decision-making process is not without merit. There are concerns that humans are driven by machine decision cycles rather than systems supporting human decision-making processes [4]. For instance, military organizations are undergoing significant transformations in technology based on the promises of AI in a number of domains [5]. However, advanced technologies have demonstrated harmful side effects in high-stakes decision-making processes. For example, accidents such as the USS Vincennes incident and the Patriot battery fratricide in Iraq [6,7,8], demonstrate the consequences of misconstruing information from automated systems.

AI will be a key component in many future technologies used by the US military, where decisions will be measured in seconds. However, decision-making may require a slowing of decision speed—something rarely addressed amidst the push for faster decision-making with AI. Studying the trust dynamics at the confluence of human factors and information features may enhance the engineering efforts of the decision environment by considering the recently improved understanding of human rationality [9,10]. For these reasons, investigating trust as an emergent phenomenon between human and AI system interactions should inform not only future engineering designs and interventions but also increase the understanding of the human response mechanisms to improve decision quality.

## 2. Background

Research on decision-making with AI is becoming increasingly important. Human decision-making is increasingly supported by AI in different domains [11]. As AI takes over more aspects of information gathering and synthesizing outputs (e.g., ChatGPT) to humans, understanding how users form trust around such outputs becomes important in decision-making processes. To support beneficial decision outcomes, humans must trust the AI outputs required for this decision-making. However, trust may be influenced by system interactions in ways that may change outcomes.

Human–AI interactions go beyond user interface (UI) and user experience (UX) designs. Recent research in areas of decision-making has uncovered additional considerations for engineering choice architectures [12,13]. For instance, asking questions or making intermediate choices in earlier stages of a decision-making process has some affect on a later stage of that same process [14]. If accurate, this may extend to other multi-stage decision-making processes with AI as well.

### 2.1. Trust in AI for Decision-Making

Researchers have examined human–AI trust dynamics in interactions through a number of controlled and simulated studies. These studies have included human–machine (AI) trust with autonomous vehicles [15,16], automated pasteurization plant operations [17], navigation tasks [18], autonomous farm equipment [19], verbal explanation models [20], and path planning [21]. The research subjects included a broad range of college students, military personnel, Amazon Mechanical Turk participants, and general community solicitations who participated in a wide variety of empirical studies. Experiments included controlled, quasi-experimental studies, field experiments, and computer simulations [22,23,24]. While a number of studies have included a broad range of interactions and environments, trust has been measured in equally diverse ways.

Trust researchers or Researchers who study trust have looked at the behavioral aspects of machines to elicit and measure trust. However, Ref. [25] found little convergence across trust measures within the empirical research. For instance, measuring trust has encompassed a broad range of techniques such as self-reported measures through surveys [26,27], correlation with biofeedback such as galvanic skin response [28], eye gaze [29,30], binary measures, such as use (trust)/disuse (distrust) [31], and behavioral measures [32,33,34]. A review of empirical research on human autonomy teams found that self-reported surveys were the most common instruments for reporting measures of trust [35]. Regarding trust in autonomy, Ref. [36] appropriately notes that”…trust is an abstraction that cannot be precisely measured. We know of no absolute trust measurement. Since trust is a relative measurement, we are restricted to measuring changes in trust” (p. 59). Hence, the body of literature on human–machine trust is diverse and eclectic. Nonetheless, how people directly interact with intelligent systems and communicate trust to teams and beyond is still a newer area of research. Still, one area that can help operationalize the concept is how trust may be measured through the act of delegation.

Thus far, there are a variety of concepts and definitions of trust in the literature. In this study, delegation is used as a surrogate for trust and is supported by the literature on trust. For instance, several researchers have explicitly stated that delegation implies trust [37,38]. Trust specifically entails reliance on another agent to accomplish something [39]. Conversely, the concept of trust inherently assumes some aspect that involves delegation to another agent for the accomplishment of some goal by the trustor [40]. Specifically related to AI, decision-making is increasingly delegated to AI in terms of trust [26,41,42,43]. In some instances, AI is making decisions without any human intervention [44]. However, this still implies delegation at a higher level of the system [43]. In [45], the authors suggest that the act of dependence overlaps with the concept of delegation and trust; thus, trust is an antecedent to delegation. For these reasons, delegation is taken as trust in action to capture trust behaviors from a human participant.

### 2.2. Quantum Probability Theory

Quantum probability theory (QPT) is increasingly being applied in decision science research. Beyond its use in physics, QPT axioms are being used to model human cognition [10,14,46,47,48,49,50,51,52]. QPT formalisms that originated from quantum theory provide a tractable means to operationalize them with decision-making concepts. Mathematical axioms of QPT ameliorate the puzzling experimental findings in physics. For instance, measurement and observation are shown to change the system under study [53]. The counterpart of this concept in decision science is that judgment/measurement *creates* [emphasis added] rather than records what existed right before the judgment [10]. The puzzling experimental findings in decision sciences have been addressed by using the axioms of the QPT. In doing so, a cognitive system is modeled with the concept of superposition. The concept of superposition supports a modeling approach that describes the evolution of a cognitive system’s states with probability amplitudes concerning all possible outcomes; hence, the system state becomes indefinite with respect to all possible outcomes and vacillates among them. This departure from a classical approach to modeling cognitive systems avoids the issue of a definite system state at any temporal moment. This requirement is a foundational assumption of the Markovian methods that follow the axioms of classical probability theory (CPT). Therefore, using QPT over traditional models of human decision-making under uncertainty has some advantages. Due to the interactive nature of human and AI decision-making, modeling approaches using QPT can improve the understanding of the dynamics and offer a more reliable foundation to engineer decision environments.

### 2.3. Modeling Human–AI Decision-Making with Quantum Probability

Human–AI decision-making must account for different types of uncertainty, which may be characterized as either epistemic or ontic. Epistemic uncertainty has to do with a lack of information and may be minimized by gathering additional information from the environment. Ontic uncertainty describes the uncertainty or indeterminacy experienced due to the superposition of the cognitive states that represent the possible outcomes [54]. Similar to polarizing a target in a high-energy particle physics experiment, posing a question frames or bounds the possible outcomes. However, the superposition of the system state is maintained, thus indeterminacy is sustained. This indeterminacy associated with ontic uncertainty can only be resolved through interaction with the environment, for example, an agent eliciting a decision. The distinctions between epistemic and ontic uncertainty are important for modeling human decision-making in situ, especially when decision-making is supported by AI. To elucidate this importance—how two perspectives (e.g., human and the perspective a human holds of the AI) can become incompatible and the relation between incompatibility and uncertainty—the following simple scenario may prove helpful.

Suppose the human and AI perspectives are represented, respectively, as *P_Human_* and *P_AI_*. If the two perspectives are commutative, $P_{Human}P_{AI}-P_{AI}P_{Human}=0$ (or very close to zero, such that it is negligible). This means that switching between the two perspectives does not impact each other or form a context for the other. On the other hand, if the two perspectives are incompatible, $P_{Human}P_{AI}-P_{AI}P_{Human}\neq 0$. In such cases, the difference is not negligible or significant and varies based on the degree of differences between the two perspectives.

In QPT, interaction is conceptualized as distinctively different from the classical understanding of an interaction. Ostensibly, the distinction emanates from an indeterminacy that enables one to capture the ambiguity that a decision-maker experiences [55,56]. The salient distinction between the classical and quantum approaches concerning the interaction is that eliciting a decision or an intermediate judgment has consequences if the involved perspectives are incompatible. Suppose *P_Human_* and *P_AI_* are two incompatible perspectives. Thinking about AI-provided information forms a significant context for a decision. Since these two perspectives are incompatible, eliciting an intermediate judgment concerning *P_AI_* will eliminate/minimize the influence, ontic uncertainty, of *P_AI_* on *P_Human_*. Similarly, asking a question (e.g., measuring or disturbing the system) can create a definite state or initiate an adaptive behavior as a result of the interaction with the system. CPT approaches, such as the Markov process, cannot model such behavior because of the definite state premise. For these reasons, the application of QPT for modeling cognitive processes shows some benefits over CPT approaches.

### 2.4. Quantum Open Systems Approach

Recent research in decision science has resulted in an alternative characterization of human decision-making with quantum-like rationality; this means that decision-making supported by machines follows more classical rationality [48,57]. Depending on situational constraints, sometimes quantum-like models dominate, and other times classical models dominate. To represent the dominating dynamics continuously, recent research suggests employing models built on a quantum open system (QOS) approach [58]. QOS can capture both the non-commutative relations between incongruent perspectives and interaction effects while also including classical Markovian dynamics. Furthermore, QOS can relate the dissipation and adaptation dynamics of decision-making with a single probability distribution as a function of time [59]. For example, in a human decision-making process supported by AI, the incompatibility of two perspectives can be modeled to include interactions of related effects over time. QOS provides a generative model of decision-making that can be used to engineer human and AI interactions over time. This dynamic modeling approach can help steer human decision-makers to better judge and reconcile differing perspectives, which can avoid both algorithmic bias and algorithmic aversion proclivities. Such an improvement in modeling decision-making behaviors could align human and AI rationalities in a way that makes them more structurally rational.

### 2.5. Trust and Ontic Uncertainty

As previously mentioned, ontic uncertainty can be resolved/minimized through measurement/observation by introducing an intermediate judgment (e.g., question–decision in a multi-stage decision process). Previous research on human trust has demonstrated that people have difficulty implementing AI solutions due to a lack of trust in AI systems. Still, trust research regarding AI typically views trust as a composite of static measures that do not capture trust as a process, which can evolve with temporal dynamics through interactions.

Research shows that human decision-making can be modeled using a quantum open system modeling that can capture human response behaviors (i.e., trust) with a single probability distribution as a function of time [59]. The dynamics of decision-making with a single continuous equation make generalization more feasible. Based on previous research [14] in the field of decision-making, it is conjectured that intermediate decisions will influence human user trust when using AI to support decision-making. To test this, we conducted the following online experiment. 

## 3. Methods

### 3.1. Participants

The participants for this study consisted of Amazon MTurk and graduate students from a West Coast graduate educational institution for military officers, totaling n = 192. In this study, the military student participants consisted of 7 females (16.3%) and 36 males (83.7%). The MTurk population consisted of 51 females (34.2%) and 98 males (65.8%). The average age of participants was 32. Of the participants, 83.7% were males; 86% identified as Caucasian; and 58.9% claimed to possess a bachelor’s degree. MTurk participants were compensated USD 15 for participating in the study.

Participants’ consent was collected prior to starting the study and included any risks associated with the study. On the consent form, participants were provided with information on why the research was being conducted. Additionally, participants were assured that the study was anonymous and that their anonymity would be preserved along with all data collected for the study. Upon completion of the study, participants were informed about the aims of the research and were given an opportunity to contact the researchers if they experienced any harm while participating in the study.

### 3.2. Overall Design

Our experiment consisted of a counterbalanced 2 × 7 factorial design (between subjects) to compare the differences between two main conditions and seven timing conditions. Participants completed a total of 21 imagery analysis tasks for screening AI-provided advice. All AI-screened images provided to participants were from a simulated AI system and not an extant AI system. This study used a simulated AI system to classify and quantify objects within a given image. This simulated AI then presented the results to a human for decision support and a subsequent delegation decision. The experimental design was conducted using Qualtrics and participants were randomly assigned to one of two main conditions (e.g., intermediate decision/no intermediate decision). In stage two of the experiment, each participant group was asked to rate on a scale from 0 to 100 (0 = not likely; 100 = very likely) how likely they were to delegate this decision to AI in the future.

### 3.3. Experimental Procedure

Participants were told that they were imagery analysts who were being assisted by a system that provided each analyst with an AI-annotated picture. An example of this is provided in Figure 1. All AI-annotated pictures contained some ambiguity that included ambiguity within the image itself or overlapping AI annotations that could obscure additional image details. In the intermediate judgment condition, participants were asked to either agree or disagree with AI’s advice as shown in the stimulus’ accompanying text. In the no-selection condition, participants were only asked to acknowledge AI-provided advice. In stage two of the experiment, all participants were asked to rate how likely they were to delegate this decision to AI. Participants were presented with a sliding scale to rate how likely they were to make this delegation decision, which ranged from 0 to 100 (0 = not likely; 100 = very likely). In the scenario, participants were informed that delegating to AI meant that AI would catalog the image as marked for faster processing in the future, and this aspect was not revisited in the experiment. Additionally, each delegation decision in the second stage included a secondary component of varying times for making that decision (e.g., 5, 10, 15, 20, 25, 30, and 35 s). Participants were not permitted to move faster than the time allotted. Varying the amount of time for making a delegation decision was elicited to probe for any temporal effects as well. Figure 2 provides a pictural illustration of the experimental procedure.

This experimentation extends previous research in the area of multistage decision-making and similarly follows several well-established experiments regarding choice selection and a subsequent capturing of the variable of interest [14,61]. This experiment extends choice (e.g., intermediate judgment) and delegation (e.g., trust) when decision-making is supported by AI. This experiment, to the best of our knowledge, was the first of its kind to capture and model trust in this way.

To focus on the concept of trust alone, several mitigations were taken to prevent participants from equating trust solely with the reliability of the AI. First, the accuracy of the AI system was calibrated at 48 per cent accuracy across all trials; however, participants were not told this accuracy, nor was an AI confidence rating provided for each stimulus. Furthermore, participants were told that the classifications came from several different AI systems but they would not know which one to prevent potential learning effects. Overall, these mitigations were taken to prevent in situ learning that could further confound reliability with trust.

## 4. Results

The delegation strength for each condition was pooled for each timing condition. A plot of these results is provided in Figure 3. Based on the results, it is clear that the choice condition (intermediate judgment) provided a boosting effect compared to the no-choice condition.

To investigate these effects further, we follow [62,63] to compute the probabilities of intermediate decisions and no intermediate decisions followed by a subsequent delegation strength rating. Table 1 summarizes the probability calculations from the experiment. To calculate a delegation decision for this experiment, the threshold value of 75 was used. This threshold assumption is based on previous research, which found that participants in other human–AI trust studies made clear trust decisions with an AI partner [64,65,66].

Based on the calculations of the joint probabilities in the choice condition compared to the no-choice condition, the researchers found clear violations of total probability between the two conditions. If implicit categorization is taking place in the no-choice condition (i.e., no intermediate judgment), then the difference between the total probability of two conditions should be ≈0. However, based on the results, it is clear that the law of total probability is violated for most of the timing conditions. The largest violations occur at the 5- and 25-s timings seen in Table 1.

To derive these predictions, a specific level of delegation strength was needed to determine trusting behavior. While it would have been easy to assume that above 50% connoted a delegation decision and interpret anything equal or below 50 as a not-delegate decision, this would have been arbitrary. First, the bimodal distribution of the data with a large percentage of observations above 55% suggests that delegation decisions may be higher than an equal odds approach. Second, other empirical research on trust behaviors with AI and robot performance had to be between 70 percent and 80 percent to elicit a trust decision from a participant [64,65,66]. For these reasons, the predictions considered 75% or above as commensurate with a delegation decision by a human to the AI. While higher deviations, e.g., violations of total probability, were found at lower thresholds (i.e., 50 per cent or 60 per cent), the researchers decided to align with extant research that suggests that trust in machines was only exhibited at higher levels of reliability, which may be equated with a delegation or trust decision.

### 4.1. Modeling Delegation Strength with Quantum Open Systems to the Study Data

The violation of total probability suggests the need for modeling techniques that can account for these violations. One theory that captures these types of violations involves quantum models of decision-making. To elucidate the difference between classical and quantum models, first, the Markov decision model is discussed, and then the quantum model.

In this model, choice outcomes for a decision-maker are to agree or disagree with a machine (i.e., AI), and decision outcomes are to delegate or not delegate. The set of choice outcome states is $Choice=\left\{\left.|A\right\rangle ,\;\left.|DisA\right\rangle \right\}$, where *A* = “agree”, and *DisA* = “disagree”. The set of decision outcome states is $Decision=\left\{\left.|D\right\rangle ,\;\left.|notD\right\rangle \right\}$, where *D* = “delegate”, and *notD* = “not delegate”. For simplicity, suppose the initial probability distribution for the agree/disagree choice is represented by a 2 × 1 probability matrix:(1)$$P_{init}=\left[\begin{array}{c} P_{A}\\ P_{DisA} \end{array}\right]$$
where $P_{A}+P_{DisA}=1$ and $P_{A},\;P_{DisA}$ are positive real numbers. In this decision process, the decision-maker starts with the probability distribution expressed in Equation (1). From the agree/disagree choice, the decision-maker transitions to delegate/not delegate states. The transition matrix that captures this behavioral process can be written as follows:(2)$$T=\left[\begin{array}{cc} T_{DA} & T_{DDisA}\\ T_{notDA} & T_{notDDisA} \end{array}\right]$$

The matrix in Equation (2) represents the four transition probabilities (e.g., $T_{DDisA}$ represents the probability of transitioning to the delegate (*D*) state from the disagree (*DisA*) state); hence, entries within each column are non-negative and the rows within each column add up to one. Then, the probability of the delegate (*D* = delegate, *notD* = not delegate) outcome can be written as follows:$$P_{final}=T\cdot P_{init}=\left[\begin{array}{c} {Pr}_{T}\left(D\right)\\ {Pr}_{T}\left(notD\right) \end{array}\right]$$
(3)$$P_{final}=\left[\begin{array}{c} P_{A}\cdot T_{DA}+P_{DisA}\cdot T_{DDisA}\\ P_{A}\cdot T_{notDA}+P_{DisA}\cdot T_{notDDisA} \end{array}\right]$$

To interpret and elucidate the final probability distribution in Equation (3), a 2 × 2 joint probability distribution table of these four events is shown in Table 2.

By using Table 2, one can write the total probability of the delegate as follows:(4)$${Pr}_{T}\left(D\right)=\mathrm{Pr}\left(D|A\right)\cdot \mathrm{Pr}\left(A\right)+\mathrm{Pr}\left(D|DisA\right)\cdot \mathrm{Pr}\left(DisA\right)$$

Subsequently, Equation (4) can be written as follows:$${Pr}_{T}\left(D\right)=\mathrm{Pr}\left(D\cap A\right)+\mathrm{Pr}\left(D\cap DisA\right)=a+b$$

Table 2 shows the decision process that elicits the agree/disagree choice of the decision-maker before deciding to delegate or not delegate. The delegate or not-delegate decision outcome can also be attained without eliciting the agree/disagree choice. The probability values for this decision process are represented in Table 3.

To use the Markov model shown in Equation (3) to capture the decision process for both the conditions shown in Table 2 and Table 3, it is necessary to assume that the condition in Equation (3) holds true. After that, to fit the data, Equation (3) requires three parameters, $P_{A}$, $T_{DA}$, and $T_{DDisA}$. These three parameters are obtained from the data, $P_{A}=\mathrm{Pr}\left(A\right)$, $T_{\mathrm{DA}}=\mathrm{Pr}\left(D|A\right)$, $T_{\mathrm{DDisA}}=\mathrm{Pr}\left(D|DisA\right)$. In return, since there are four data points ($\mathrm{Pr}\left(A\right)$, $\mathrm{Pr}\left(D|A\right)$, $\mathrm{Pr}\left(D|DisA\right)$, and $\mathrm{Pr}\left(D\right)$), one degree of freedom remains to test the model. This degree of freedom is imposed by the law of total probability that the Markov model must obey. Thus, the Markov model requires (and must predict) that ${Pr}_{T}$ (from Table 1) = Pr(D) (from Table 2). In the case where ${Pr}_{T}$ (from Table 1) ≠ $\mathrm{Pr}\left(D\right)$ (from Table 2), the Markov model becomes applicable if the transition matrix entries change; in this case, the model cannot be empirically tested [10,62].

Similar to the Markov model, the quantum model of choice states are $Choice=\left\{\left.|A\right\rangle ,\;\left.|DisA\right\rangle \;\right\}$ and the set of decision outcome states is $Decision=\left\{\left.|D\right\rangle ,\;\left.|notD\right\rangle \right\}$. However, different than the Markov model, in a quantum model, probabilities are replaced by amplitudes. The initial amplitude values for the agree/disagree choice are represented by a 2 × 1 matrix:(5)$$\Phi_{init}=\left[\begin{array}{c} \phi_{A}\\ \phi_{DisA} \end{array}\right]$$

The probability of observing an ‘agree’ choice becomes $\mathrm{Pr}\left(A\right)={\left|\phi_{A}\right|}^{2}$; the probability of observing the ‘disagree’ choice becomes $\mathrm{Pr}\left(DisA\right)={\left|\phi_{DisA}\right|}^{2}$. The sum of the squared amplitude equals one, ${\left|\phi_{A}\right|}^{2}+{\left|\phi_{DisA}\right|}^{2}=1$.

In the case of the quantum model, choice and decision outcomes form two orthogonal bases in a two-dimensional Hilbert space. Peculiar to the quantum model, the events describing the choice–agree basis can be incompatible with the delegation decision basis, and used to represent system states as follows:(6)$$\left.|S\right\rangle =\phi_{A}\left.|A\right\rangle +\phi_{DisA}\left.|DisA\right\rangle =\phi_{D}\left.|D\right\rangle +\phi_{notD}\left.|notD\right\rangle$$

When a stimulus is presented to a decision-maker, the cues in the situation generate a superposition state with respect to agree and disagree, and delegate and not delegate. By asking a choice question, agree–disagree, the system state, shown in Equation (6), becomes a superposition of $\left.\left|A\right.\right\rangle$ and $\left.\left|DisA\right.\right\rangle$. To explicate the concept of collapse, consider the case of the deeply virtual Compton scattering experiment [67,68]. In this experiment, one goal is to hit one quark with the incoming electron beam. The key to this experiment’s success is the polarization of the target, which can be He. By polarizing the target, the amplitudes in the wave function equation can be aligned with the beam polarization. However, the polarization of the target does not mean that a collapse happens until the beam and the target interact. In the context of this paper, asking questions is considered polarization. By asking a question, a new set of cues is introduced; subsequently, the equation that represents the cognitive system of the decision-maker is primed toward the possible outcomes for that question. Since the outcomes are still associated with possibilities, asking questions changes the amplitudes, making the set of answers to the question more possible. As shown in Figure 4 and Figure 5, when a decision-maker chooses to agree (resulting in agree then delegate, as shown in Figure 4) or disagree (resulting in disagree and delegate, as shown in Figure 5), the superposition of states with respect to the agree–disagree basis is resolved. Figure 5 corresponds to the first element ($\mathrm{Pr}\left(D|A\right)\cdot \mathrm{Pr}\left(A\right)$) of the total probability shown in Equation (4); Figure 6 represents the second element ($\mathrm{Pr}\left(D|DisA\right)\cdot \mathrm{Pr}\left(DisA\right)$) of Equation (4). Subsequently, either a delegate or not-delegate state is chosen from a new superposition of states, with the delegate or not-delegate states. Contrary to this process as shown in Figure 6, if the agree–disagree question is never asked, a decision-maker chooses delegate or not delegate (without expressing the agree or disagree choice), and the decision-maker never resolves the superposition concerning the agree/disagree basis. This is one of the salient differences between quantum and Markov models.

Another difference between the quantum and Markov models involves the calculation of the transition matrix. In the quantum model, the transition amplitudes are represented as the elements of a unitary matrix:(7)$$U=\left[\begin{array}{cc} u_{DA} & u_{DDisA}\\ u_{notDA} & u_{notDDisA} \end{array}\right]$$ Since the matrix in (7) is a unitary matrix, it must satisfy the following conditions:(8)$$U^{†}\cdot U=U\cdot U^{†}=I$$ The requirements in Equation (8) also imply the following:$$\begin{array}{c} {|u}_{DA}{|}^{2}+{|u}_{notDA}{|}^{2}=1\\ \begin{array}{c} {|u}_{DDisA}{|}^{2}+{|u}_{notDDisA}{|}^{2}=1\\ \begin{array}{c} {|u}_{DA}{|}^{2}+{|u}_{DDisA}{|}^{2}=1\\ \begin{array}{c} {|u}_{notDA}{|}^{2}+{|u}_{notDDisA}{|}^{2}=1\\ \begin{array}{c} u_{DDisA}^{*}\cdot u_{DA}+u_{notDDisA}^{*}\cdot u_{notDA}=0\\ u_{DA}\cdot u_{notDA}^{*}+u_{DDisA}\cdot u_{notDDisA}=0 \end{array} \end{array} \end{array} \end{array} \end{array}$$

Similar to a Markov model, a transition matrix is generated in the quantum model. In this case, the elements of the transition matrix are generated from the unitary matrix. The resulting transition matrix is doubly stochastic, with each of its rows and columns summing to unity. For the choice condition, transition probabilities are calculated by the squared magnitudes of the unitary matrix elements:$$T_{ij}\left(U_{ij}\right)={\left|U_{ij}\right|}^{2}$$
where $T_{ij}$ represents the transition matrix; hence, $T\left(U\right)$ must be doubly stochastic. A decision-only situation, directly deciding to delegate or not delegate, is modeled by the following matrix product:(9)$$\Phi_{final}=U\cdot \Phi_{init}=\left[\begin{array}{c} \phi_{A}\cdot u_{DA}+\phi_{DisA}\cdot u_{DDisA}\\ \phi_{A}\cdot u_{notDA}+\phi_{DisA}\cdot u_{notDDisA} \end{array}\right]$$ Solving Equation (9) results in the following:$$\Phi_{D}={(\phi }_{A}\cdot u_{DA}+\phi_{DisA}\cdot u_{DDisA})\cdot (\phi_{A}\cdot u_{DA}+\phi_{DisA}\cdot u_{DDisA}{)}^{*}$$ After expanding the complex conjugate, Equation (9) is as follows:(10)$$\Phi_{D}={\left|\phi_{A}\right|}^{2}{\left|u_{DA}\right|}^{2}+{|\phi }_{DisA}{|}^{2}{\left|u_{DDisA}\right|}^{2}+\;\underset{Interference\;term\;for\;delegate\;=\;Int_{D}}{\underbrace{2\cdot \left|\phi_{DisA}\right|\cdot \left|u_{DDisA}\right|\cdot \left|\phi_{A}\right|\cdot \left|u_{DA}\right|\cdot \cos\;\left(\theta \right)\;\;\;}}\;$$
where the θ term in Equation (10) is the phase of the complex number ${(\phi }_{A}\cdot u_{DA})\cdot {(\phi }_{DisA}\cdot u_{DDisA}{)}^{*}$; in Equation (10), only the real part of the complex number is used $Int_{D}=2\cdot Re\left[{(\phi }_{A}\cdot u_{DA})\cdot {(\phi }_{DisA}\cdot u_{DDisA}{)}^{*}\right]$.
(11)$$\Phi_{D}=\underset{total\;probability\;Equation\;(4)}{\underbrace{p_{A}T_{DA}+p_{DisA}T_{DDisA}}}+Int_{D}$$

Equation (10) is called the law of total amplitude [10,62]. As can be seen in Equation (11), because of the interference term, Equation (10) violates the law of total probability. Depending on the value of cos θ, the probability produced by Equation (10) can be higher than that of Equation (4) or less. In the case where cos θ=0, the interference term becomes zero in Equation (10). Following the discussion in [10,62], we proceed with the four-dimensional model because the two-dimensional model of this decision-making process demonstrates the same violation of the double stochasticity requirement for the quantum model.

Capitalizing on the state concepts in two-dimensional models, the four-state model will include the following combination of decision states:(12)$$S=|\left.A,D\right\rangle ,|\left.A,notD\right\rangle ,|\left.DisA,D\right\rangle ,\left.|DisA,notD\right\rangle$$

The state $|\left.A,D\right\rangle$ in Equation (12) represents the state where the decision-maker agrees with AI and delegates the decision to the AI. Due to the dynamics of the Markov model, even though the state of the system is not known by the modeler, the system will be in a definite state and jump from one to another state or stay in the same state. The initial probability distribution of this four-dimensional model is represented by a 4 × 1 matrix, as follows:(13)$$P_{init}\left(0\right)=\left[\begin{array}{c} p_{AD}\left(0\right)\\ \begin{array}{c} p_{AnotD}\left(0\right)\\ \begin{array}{c} p_{DisAD}\left(0\right)\\ p_{DisAnotD}\left(0\right) \end{array} \end{array} \end{array}\right]$$

Each row in Equation (13) represents the probability of being in any of the states listed in Equation (12) at time, t=0; for example, $p_{AD}\left(0\right)$ is the probability of ‘agree’ and ‘delegate’ at t=0.

For the process where the choice (agree/disagree) precedes the (delegate/not delegate) task, the condition of choosing ‘agree’ would require having zero values for the third ($p_{DisAD}\left(0\right)=0$) and fourth ($p_{DisAnotD}\left(0\right)=0$) entries of the matrix. In return, $p_{AD}\left(0\right)+p_{AnotD}\left(0\right)=1$ because choosing ‘agree’ allows only these two states as probable outcomes. As a result, the initial probability distribution, $P_{init}=P_{A}$.

In the decide-only task, for the Markov model, it is assumed that both agree and disagree are probable, but these probability values are not known. Then, by capitalizing on the discussion in [62], for this task, the initial probability distribution is as follows:(14)$$P_{init(decision\;alone)}=p_{A}P_{A}+p_{DisA}P_{DisA}$$
where $p_{A}=1-p_{DisA}$, which represents the implicit probability of ‘agree’ for the decision-alone task.

The state evolution for the choice condition is as follows. After choosing to agree/disagree, a decision-maker decides to delegate/not delegate. The decision can take place anytime, *t,* after the agree/disagree choice. The cognitive process that represents the state evolution of the decision-maker during time t−0 can be represented by a 4 × 4 transition matrix, $T_{ij}\left(t\right)$. This transition matrix represents transitioning probabilities from state *i* to state *j*. The time-dependent probability distribution across all of the states in Equation (12) can be expressed as follows:(15)$$P_{final}\left(t\right)=T_{ij}\left(t\right)P_{init}$$
(16)$$P_{final}\left(t\right)=\left[\begin{array}{c} P_{AD}\left(t\right)\\ \begin{array}{c} P_{AnotD}\left(t\right)\\ \begin{array}{c} P_{DisAD}\left(t\right)\\ P_{DisAnotD}\left(t\right) \end{array} \end{array} \end{array}\right]$$ Then, at any time, *t*, the probability of delegating can be expressed as follows:(17)$$P_{AD}\left(t\right)+P_{DisAD}\left(t\right)$$

The transition matrix for any Markov model must satisfy the Chapman–Kolmogorov equation, and the solution of this transition matrix results in the following:(18)$$\frac{dT\left(t\right)}{dt}=K\cdot \left(t\right)\to T\left(t\right)=e^{K\cdot t}$$
where *K* is the intensity matrix with non-negative off-diagonal entries and the rows within each column sum to zero, which is required to generate a transition matrix for the Markov model. Typically, the mental processes concerning agree/disagree and delegate/not delegate are captured with the intensity matrix.

Following the discussion in [62], defining an indicator matrix is required to operationalize and link Equations (17) and (18) to the choice and decision tasks. The indicator matrix for these two tasks will be a 4 × 4 matrix, as follows:(19)$$M_{D}=\left[\begin{array}{c} 1\\ \begin{array}{c} 0\\ \begin{array}{c} 0\\ \underset{|\left.AD\right\rangle }{\underbrace{0}} \end{array} \end{array} \end{array}\begin{array}{c} 0\\ \begin{array}{c} 0\\ \begin{array}{c} 0\\ \underset{|\left.AnotD\right\rangle }{\underbrace{0}} \end{array} \end{array} \end{array}\begin{array}{c} 0\\ \begin{array}{c} 0\\ \begin{array}{c} 1\\ \underset{|\left.DisAD\right\rangle }{\underbrace{0}} \end{array} \end{array} \end{array}\begin{array}{c} 0\\ \begin{array}{c} 0\\ \begin{array}{c} 0\\ \underset{|\left.DisAnotD\right\rangle }{\underbrace{0}} \end{array} \end{array} \end{array}\right]$$

The matrix in Equation (19) ensures that only the delegate events are included in the matrix multiplication of $M_{D}$ and $P_{final}$; to calculate Equation (15) from $M_{D}\cdot P_{final}$, a 1 × 4-row matrix is necessary to calculate the probability of the delegate as follows:$$L=\left[1\;1\;1\;1\right]$$ By using the *L* matrix, the final probability for ‘agree’ and the delegate ($|\left.AD\right\rangle$) is as follows:(20)$$\mathrm{Pr}\left(D|A\right)=L\cdot \left(M_{D}\cdot T\left(t\right)\cdot P_{A}\right)$$

To complete Equation (20) by using the *L* matrix, the final probability for ‘disagree’ and the delegate ($|Dis\left.AD\right\rangle$) is as follows:(21)$$\mathrm{Pr}\left(D|DisA\right)=L\cdot \left(M_{D}\cdot T\left(t\right)\cdot P_{DisA}\right)$$ For the decision-only task, the probability of the delegate is expressed as follows:(22)$$\mathrm{Pr}\left(A\right)=L\cdot M_{D}\cdot T\left(t\right)\cdot \left(P_{A}\cdot P_{A}+P_{DisA}\cdot P_{DisA}\right)$$
(23)$$\mathrm{Pr}\left(A\right)=P_{A}\cdot \underset{Equation\;(20)}{\underbrace{L\cdot M_{D}\cdot T\left(t\right)}}+P_{A}\cdot p_{DisA}\cdot L\cdot \underset{Equation\;(21)}{\underbrace{M_{D}\cdot T\left(t\right)\cdot P_{DisA}}}$$
(24)$$\mathrm{Pr}\left(A\right)=P\cdot \mathrm{Pr}\left(D|A\right)+p_{DisA}\cdot \mathrm{Pr}\left(D|DisA\right)$$

In this study, by using previous research methods in references [62,69], the values of implicit probabilities $p_{A}\;and\;p_{DisA}$ are determined by $p_{A}=\mathrm{Pr}\left(A\right)$ and $p_{DisA}=\mathrm{Pr}\left(DisA\right)$. $\mathrm{Pr}\left(A\right)$ and $\mathrm{Pr}\left(DisA\right)$ are observed probabilities from categorization tasks. Although this might involve the subjective determination of these values, the Markov model in Equation (24) becomes the weighted average of $\mathrm{Pr}\left(D|A\right)$ and $\mathrm{Pr}\left(D|DisA\right)$ and will not match the agree/disagree and delegate condition probabilities [62].

Identical to four-dimensional Markov model states, as shown in Equation (12), the four-dimensional quantum model has four decision states, as follows:(25)$$S=|\left.A,D\right\rangle ,|\left.A,notD\right\rangle ,|\left.DisA,D\right\rangle ,\left.|DisA,notD\right\rangle$$

The state $|\left.A,D\right\rangle$ in Equation (25) represents the state where the decision-maker agrees with AI and will delegate the decision to the AI. Due to the nature of the quantum model, the initial probability distribution is in a superposition of all of the states shown in Equation (25). The initial probability distribution of this model is represented by a 4 × 1 column vector, as follows:(26)$$\Phi_{init}=\left[\begin{array}{c} \phi_{AD}\\ \begin{array}{c} \phi_{AnotD}\\ \begin{array}{c} \phi_{DisAD}\\ \phi_{DisAnotD} \end{array} \end{array} \end{array}\right]$$

The elements of Equation (26) represent the probability amplitudes (not transition amplitudes), which are complex numbers for each of the states in Equation (26), and the sum of the squared amplitudes of these elements is one:(27)$${\left|\phi_{AD}\right|}^{2}+{\left|\phi_{AnotD}\right|}^{2}+{\left|\phi_{DisAD}\right|}^{2}+\;{\left|\phi_{DisAnotD}\right|}^{2}={\left|\Phi_{init}\right|}^{2}=1$$

Similar to the Markov model, these probability amplitudes vary with the experimental task.

For the task in which the choice is to agree/disagree and the decision is to delegate/not delegate, if the choice equals ‘agree’, then ${\left|\phi_{AD}\left(0\right)\right|}^{2}+{\left|\phi_{AnotD}\left(0\right)\right|}^{2}=1$. As a result, the initial amplitude distribution is $\Phi_{init}=\Phi_{A}$. The foundational difference between the Markov and quantum models is distinguishable for the second task, which is the decision-only (delegate/not delegate) condition. In this condition, according to the quantum model, a decision-maker never resolves his/her superposition of states concerning agree/disagree; hence, the initial amplitude distribution is as follows:(28)$$\Phi_{init}=\left(\phi_{A}\cdot \Phi_{A}+\phi_{DisA}\cdot \Phi_{DisA}\right)$$

As in the Markov model, after choosing to agree or disagree, the decision-maker makes a decision at some period of time, *t*. To represent the cognitive processes of deliberation between choosing to agree/disagree at time *t*, the 4 × 4 unitary matrix ($U\left(t\right)$) is used. This $U\left(t\right)$ updates the superposition of the initial amplitude distribution:(29)$$\Phi_{final}=U\left(t\right)\cdot \Phi_{init}$$
where $U{\left(t\right)}^{†}\cdot U\left(t\right)=I$ ensures the preservation of the inner products, and ${\left|U\left(t\right)\right|}^{2}=T\left(t\right)$, which is the transition probability matrix. For example, with $U_{ij}\left(t\right)$ denoting the unitary matrix, the transition probability from state i to j equals the following:(30)$$T_{ij}\left(t\right)={\left|U_{ij}\left(t\right)\right|}^{2}$$

The transition matrix in (30) must be doubly stochastic. As discussed in [62], the transition matrix for the quantum model satisfies the Chapman–Kolmogorov equation, $U\left(t+\Delta t\right)=U\left(t\right)\cdot U\left(\Delta t\right)$; therefore, the unitary matrix, $U\left(t\right)$, satisfies the following equation:(31)$$\frac{dU\left(t\right)}{dt}=-i\cdot H\cdot U\left(t\right)$$
where H is the Hermitian Hamiltonian matrix. The solution to Equation (31) is as follows:(32)$$U\left(t\right)=e^{-i\cdot H\cdot t}$$

Equation (32) is a matrix exponential function, and it allows the construction of a unitary matrix at any point in time with the same Hamiltonian.

Equation (29) represents the amplitude distribution at any time, *t*, and can be expressed as follows:(33)$$\Phi_{final}\left(t\right)=\left[\begin{array}{c} \phi_{AD}\left(t\right)\\ \begin{array}{c} \phi_{AnotD}\left(t\right)\\ \begin{array}{c} \phi_{DisAD}\left(t\right)\\ \phi_{DisAnotD}\left(t\right) \end{array} \end{array} \end{array}\right]$$

By using Equation (33), the probability of the delegate can be expressed as follows:(34)$${\left|\phi_{AD}\left(t\right)\right|}^{2}+{\left|\phi_{DisAD}\left(t\right)\right|}^{2}$$

To represent the probability values, as defined in the Markov model, a 4 × 4 matrix is defined for the quantum model as follows:(35)$$M_{D}=\left[\begin{array}{c} 1\\ \begin{array}{c} 0\\ \begin{array}{c} 0\\ 0 \end{array} \end{array} \end{array}\begin{array}{c} 0\\ \begin{array}{c} 0\\ \begin{array}{c} 0\\ 0 \end{array} \end{array} \end{array}\begin{array}{c} 0\\ \begin{array}{c} 0\\ \begin{array}{c} 1\\ 0 \end{array} \end{array} \end{array}\begin{array}{c} 0\\ \begin{array}{c} 0\\ \begin{array}{c} 0\\ 0 \end{array} \end{array} \end{array}\right]$$

Multiplying $\Phi_{final}$ by $M_{D}$ results in a vector ($\Phi_{final}\cdot M_{D}$), which includes amplitudes for the delegate and then the agree and disagree cases. As a result, the probability of delegation is as follows:(36)$${\left|\Phi_{final}\cdot M_{D}\right|}^{2}$$

Following this discussion, the probability values of the delegate for the agree and disagree conditions are as follows:(37)$$\Phi_{D|A}=M_{D}\cdot U\left(t\right)\cdot \Phi_{A}\;\mathrm{Pr}\left(D|A\right)={|\Phi }_{D|A}{|}^{2}={|M}_{D}\cdot U\left(t\right)\cdot \Phi_{A}{|}^{2}$$
(38)$$\Phi_{D|DisA}=M_{D}\cdot U\left(t\right)\cdot \Phi_{DisA}\;\mathrm{Pr}\left(D|DisA\right)={|\Phi }_{D|DisA}{|}^{2}={|M}_{D}\cdot U\left(t\right)\cdot \Phi_{DisA}{|}^{2}$$

In the condition that comprises a decision-only condition, the probability of the delegate is as follows:$$\mathrm{Pr}\left(D\right)={\left|M_{D}\cdot U\left(t\right)\cdot \left(\phi_{A}\cdot \Phi_{A}+\phi_{DisA}\cdot \Phi_{DisA}\right)\right|}^{2}$$
$$\mathrm{Pr}\left(D\right)={\left|{\phi_{A}\cdot (M}_{D}\cdot U\left(t\right)\cdot \Phi_{A})+\phi_{DisA}\cdot \;\left(M_{D}\cdot U\left(t\right)\cdot \Phi_{DisA}\right)\right|}^{2}$$
$$\mathrm{Pr}\left(D\right)={\left|\phi_{A}\cdot \Phi_{D|A}\right|}^{2}+{\left|\phi_{DisA}\cdot \Phi_{D|DisA}\right|}^{2}+2\cdot \left|\phi_{A}\cdot \phi_{DisA}\cdot \left(\Phi_{D|A}^{†}\cdot \Phi_{D|DisA}\right)\right|\cdot \cos\left(\theta \right)$$
(39)$$\mathrm{Pr}\left(D\right)=p_{A}\cdot \mathrm{Pr}\left(D|A\right)+p_{DisA}\cdot \mathrm{Pr}\left(D|DisA\right)+\;2\cdot \left|\begin{array}{c} \phi_{A}\cdot \;\;\;\;\;\;\;\;\;\;\;\;\;\;\;\\ \;\;\;\;\;\;\;\;\;\;\phi_{DisA}\cdot \left(\Phi_{D|A}^{†}\cdot \Phi_{D|DisA}\right) \end{array}\right|\cdot \cos\left(\theta \right)\;$$
where θ is the phase angle of the complex number $\phi_{A}\cdot \phi_{DisA}\cdot \left(\Phi_{D|A}^{†}\cdot \Phi_{D|DisA}\right)$. As can be seen in Equation (39), the total probability is violated when $\cos\left(\theta \right)\neq 0$.

### 4.2. Comparison between Markov and Quantum Models

Any decision task involving multiple agents (human and machine or human and human), conflicting or inconsistent information for identifying a target, and delegating a decision to the other agent involves multiple cognitive processes and their state evolution. Incoming information can often be uncertain or inconsistent, and in some instances, a decision must still be made as quickly as possible. However, it is often impractical to reprocess and resample data, and such work may result in missing a critical temporal decision window.

Decision tasks and their conceptual peripheries accentuate the importance of trust in decision-making. A decision theory used in this context can provide the probability of making a choice and is typically conducted by assigning a trust rating to decision outcomes and analyzing the distribution of time taken to decide or choose. For instance, random walks are commonly used to model these types of decision tasks. In fact, Markov models and quantum random walks are among these models. Random walk models are used quite often in the field of cognitive psychology [61] and are a good way to model multi-agent decision-making situations. These models are additionally beneficial because when the stimulus is presented, a decision-maker samples evidence from the “source” at each point in time. The sampled evidence/information changes the trust regarding the decision outcomes (e.g., delegate or not delegate). Trust may increase or decrease depending on the adjacent confidence level and the consistency of the sampled-out information to include the trustworthiness of the source. This switching between states continues until a probabilistic threshold (intrinsic to the decision-maker) is reached to engender a delegate or not delegate decision. In this context, trust continuously influences the time evolution of the system state as it transitions from one state to another, as shown in Figure 7. This influence can be captured by the intensity matrix for the Markov model, or by the Hamiltonian for the quantum model.

As an illustration, a Markov model and a quantum model were used to describe the state transitions. In the case of a nine-state model, using a Markov model requires that the decision-maker be at one of the nine states at any time (even if the modeler does not know that state, as shown in Figure 7). The initial probability distribution concerning the states for the Markov model will, thus, be $\phi_{state}=1/9$; consequently, the system starts on one of these nine states. On the other hand, using a quantum model, the nine states are represented with nine orthonormal bases in a nine-dimensional Hilbert space. Another key difference of the quantum model is that if there is no definite state at any time, t, the system will be in a superposition state, and the initial distribution is also a superposition of the nine states, as seen in Figure 8. Therefore, instead of a probability distribution, there is an amplitude distribution with equal amplitude values, $\psi_{state}=\frac{1}{\sqrt{9}}$.

In addition to the initial state distribution and evolution of the system state, jumping from one state to another vs. evolution as a superposition state, Markov models must obey the law of total probability, and quantum models obey the law of double stochasticity. Due to the nature of the Markov model, the law of total probability and jumping from one definite state to another generates a definite accumulating trajectory for the evolution of the delegation rate, which is influenced by the inconsistencies of the information, evidence, and trust of the source. On the other hand, the quantum model starts in a state of superposition and evolves as a superposition of states across time for the duration of the tasks.

As can be seen in Figure 9 (nine-state case), the Markov model predicts a gradual increase in the delegation rate, subsequently reaching an equilibrium state at around 3.5 (which could mean that the state jumps between 3 and 4). As discussed in [10], the probability distribution across states for the Markov model behaves like sand blown by the wind. The sand pile begins with a uniform distribution, but as the wind blows, it piles up against a wall on the right-hand side of the graph, which is analogous to evidence accumulation. As more sand piles up on the right, the delegation rate becomes trapped between a certain state, which is the equilibrium state.

As can be seen in Figure 9, the quantum model predicts that the delegation rate initially increases and then begins oscillating around an average value of 1.1; however, there is no definite state for this distribution. As analogized in [10], the quantum model behaves like water blown by the wind. The water is initially distributed equally across states, but when a wind begins blowing, the water is pushed against the wall on the right-hand side of the graph and then recoils back to the left side of the graph; hence, the oscillatory behavior emerges. In the context of trust, these two behaviors can capture two unique aspects of trust. The Markov model can represent a case in which trust pushes the decision-maker to a decision in favor of AI; or in the no-trust case, the decision-maker is pushed to a decision that is not in favor of AI. However, real-time decision-making involves hesitation that results in vacillation for the decision-maker. The dynamics in Figure 9 are illustrative only; concrete definitions of the intensity matrix that represent Markov dynamics and quantum dynamics are given below in the quantum open systems section.

## 5. Quantum Open Systems

The quantum open systems approach to modeling combines the Markov and quantum models into a single equation. With a single equation, Markov and quantum models may be viewed as two ends of a spectrum for different random walks [59]. This allows the modeler to tune the equation in a way that can best capture the exhibited system dynamics. Figure 10 shows the behavior of a quantum open system within an Observe, Orient, Decide, and Act (OODA) decision making system [4]. The system starts in a quantum state, which is demonstrated by the oscillatory behavior. Over time, the system is perturbed through an interaction or measurement that transitions the system into a Markov state where the behavior switches to a monotonic increase/decrease. Empirical research has also demonstrated support for using quantum open systems by highlighting them as comprehensive systems that can account for evolution, oscillation, and choice-induced changes [70].

The application of the quantum open system to decision-making with AI is applied for several reasons. First, decision-makers are not isolated systems. Hence, decision-makers interact with the other agents and in environments where reciprocal dynamics may ensue. Similarly, quantum systems are never isolated in nature and similarly, and interactions create random fluctuations within the mind of a decision-maker, evolving from a pure quantum state to a mixed state without the superposition effects [71]. Based on previous categorization and subsequent decision-making research [14,59,63,72,73,74], the application of quantum open systems appears to best describe human interaction with an AI, as eliciting an agreement with the AI is different from not eliciting a preference about the AI. Moreover, categorization and subsequent decision-making with AI and automation are found in a variety of real-world instances in the literature, such as target identification (classification) and decision-making (target engagement) [3,75,76]. Second, conventional decision-making frameworks such as recognition-primed decision models [77], and situational awareness [78], have face validity but lack empirical support for understanding how categorization affects subsequent judgments in decision-making [74]. System 1 and System 2 thinking [79], and subsequent research in this line, have empirical support but lack theoretical support. Third, the quantum open system considers both ontic and epistemic types of uncertainty experienced by human decision-makers. Including both types of uncertainty allows for a better approximation of the decision-making process as it evolves as a superposition of possible states. Moreover, it allows the quantum open system equation to capture the resolution of uncertainty and describes the temporal evolution of the mental state of the decision-maker [71]. Equally important, the quantum open system equation can model more comprehensive probability distributions because it does not depend on a definite initial system state [71].

The quantum open system provides more than a weighted average of Markov and quantum dynamics, but the integration of both provides a single probability distribution [80]. As a result, the need to switch between two methods can continuously be achieved with a single equation. Moreover, the equation allows for the time-evolved modeling of a decision with interactive components that can potentially result in a multifinality of probability distributions. Lastly, the quantum open system provides a mathematically rigorous explanation of Human-in-the-loop (HITL) behaviors. Such a mathematical explication provides a kind of epistemic affordance to the understanding of decision-making, particularly for the situation in which harnessing AI systems requires mathematical formalisms for modeling human behavior [81,82]. As a result, the use of quantum open systems provides a number of novel ways for modeling HITL-AI decision-making.

### 5.1. Quantum Open System Equation Components and Explanations

The quantum open system equation has a long history outside of its application in social and information science applications. The quantum open system is an extension of the Gorini–Kossakowski–Sudarshan–Lindblad (GKSL) equation, often shortened to the Lindblad equation [83,84]. The quantum open system is composed of a number of different parts. The equations provided in (40) through (43) are found in [14], where they were used to model temporal judgment and constructed preference.
(40)$$\frac{d}{dt}\rho \left(t\right)=-i\cdot \left(1-\alpha \right)\cdot \left[H,\rho \right]+\alpha \cdot L\left(\rho \right)$$
(41)$$L\left(\rho \right)=\sum \gamma_{ij}\cdot \left(L_{ij}\cdot \rho \cdot L_{ij}^{†}-\frac{1}{2}\left\{L_{ij}^{†}\cdot L_{ij,},\rho \right\}\right)$$
(42)$$\left\{\left(L_{ij}^{†}\cdot L_{ij,}\right),\rho \right\}=\left(L_{ij}^{†}\cdot L_{ij,}\right)\cdot \rho +\rho \cdot \left(L_{ij}^{†}\cdot L_{ij,}\right)$$
(43)$$\left[H,\rho \right]=H\cdot \rho -\rho \cdot H$$

The open system in Equation (40) is used to describe cognition and human decision-making with various applications [14,80,85]. The first part of Equation (40), $i\left[H,\rho \right]$), represents the quantum component. The second part of Equation (40), L(ρ)), represents the classical Markov component. The weighting parameter, α, in Equation (40) provides a means to weigh which element (e.g., Markov or quantum) will dominate the system. For example, when α=0, it signifies that the system is a fully quantum regime which indicates a higher ontic uncertainty. Conversely, when α=1, quantum dynamics no longer take place and the system model becomes Markovian. The quantum open system models begin with oscillatory behavior, and due to the interaction with the environment, the oscillation dissipates to a steady state, as $\underset{t\to \infty }{\lim}\hat{\rho }\left(t\right)={\hat{\rho }}_{steady}$ [83].

Different from the two- and four-dimensional quantum and the Markov models, state representation is modeled by a density matrix, represented by ρ. In Equation (40), a density matrix that is an outer product of the system state is represented by the following:(44)$$\rho =\psi \cdot \psi^{†}$$

Using a density operator provides a unique advantage to model human cognition and the decision process because a single probability distribution is used in both the quantum and Markov components [59]. In other words, this description is the evolution of the superposition of all possible states and it is a temporal oscillation. The temporal oscillation (i.e., superposition state) can vanish transiently due to a measurement or interaction with another agent or environment. Pure quantum models cannot describe a system when the system becomes an open system, which means it is no longer isolated. When a system starts interacting with the environment, the superposition state of the system begins dissipating, which is called decoherence. Decoherence is the transition from a pure state to a classical state, particularly when there is interaction from an environment [10,86]. The concept of decoherence is, however, controversial [71]; therefore, its interpretation is not discussed here. Pure Markov models, on the other hand, demonstrate accumulative behavior, failing to capture indecision represented by oscillatory behavior.

To demonstrate the evolution of the density operators, Equation (44) can be written as follows:(45)$$\rho \left(t\right)=\left.\left|\psi \left(t\right)\right.\right\rangle \left\langle \left.\psi \left(t\right)\right|\right.$$ The elements of a density operator can be written as follows:(46)$$\left.|\psi \right\rangle =\sum_{n}c_{n}\left.|n\right\rangle$$ Then, a density operator can be written as follows:$$\rho =\left.|\psi \right\rangle \left\langle \psi |\right.=\left(\sum_{n}c_{n}\left.|n\right\rangle \right)\left(\sum_{m}c_{m}^{*}\left\langle m\right.|\right)=\;\underset{diagonal\;terms}{\underbrace{\sum_{n}{\left|c_{n}\right|}^{2}|\left.n\right\rangle \left\langle n\right.|}}+\underset{off\text{-}diagonal\;terms}{\underbrace{\sum_{m\neq n}c_{n}c_{m}^{*}\left.|n\right\rangle \left\langle m|\right.}}$$

By using Equation (44), a two-state system, $\left.|\psi \right\rangle =\xi \left.|0\right\rangle +\beta |\left.1\right\rangle$, can be represented with a density matrix, as follows:(47)$$\left.|\psi \right\rangle \left\langle \psi |\right.=\underset{diagonal}{\underbrace{{\left|\xi \right|}^{2}\left.|0\right\rangle \left\langle 0|\right.+{\left|\beta \right|}^{2}\left.|1\right\rangle \left\langle 1|\right.}}+\;\underset{off\text{-}diagonal\;terms}{\underbrace{\xi \beta^{*}\left.|0\right\rangle \left\langle 1|\right.+\xi^{*}\beta \left.|1\right\rangle \left\langle 0|\right.}}$$

The matrix representation of Equation (47) with the state vectors |0=10 and |1=01 can be written as follows:(48)$$\;\left(\begin{array}{cc} {\left|\xi \right|}^{2} & \xi^{*}\beta \\ \xi \beta^{*} & {\left|\beta \right|}^{*} \end{array}\right)\;$$

When a system becomes perturbed, an interaction with the environment begins. By using the matrix representation in (48), the transition from a quantum state to a classical state can be represented as follows:(49)$$\underset{Quantum\;State}{\underbrace{\left(\begin{array}{cc} {\left|\xi \right|}^{2} & \xi^{*}\beta \\ \xi \beta^{*} & {\left|\beta \right|}^{*} \end{array}\right)}}\;\vec{interaction}\;\underset{Classical\;Ensamble}{\underbrace{\left(\begin{array}{cc} {\left|\alpha \right|}^{2} & 0\\ 0 & {\left|\beta \right|}^{2} \end{array}\right)}}$$

Examining the evolution of Equation (49) provides a comprehensive understanding of the system’s behavior. Such a full picture can be captured by the Lindblad equation shown in Equation (40), which provides a more general state representation of the system, in this case, a cognitive system, by including a probability mixture across pure states:(50)$$\rho \left(t\right)=\sum_{j}p_{j}\cdot \psi \cdot \psi^{†}$$

As discussed in [80], through linearity, the density matrix in Equation (50) follows the same quantum temporal evolution as in Equation (40). Consequently, the density matrix captures two types of uncertainty, epistemic and ontic. Epistemic uncertainty represents an observer’s (e.g., modeler’s) uncertainty about the state of the decision-maker. An ontic type of uncertainty represents the decision-maker’s indecisiveness or internal uncertainty concerning the stimuli via the superposition of possible states. In other words, a decision-maker’s ambiguity over evidence may be described as the vacillation of a decision-maker.

For the Hamiltonian matrix (51), the diagonal elements, or drift parameters, $\mu_{Q},$ control the rate at which the quantum component of the system state (superposition) shifts to an alternate delegation strength. These elements are known as the potential that captures the dynamics pulling the superposition state back to a higher amplitude of a basis state in a specific column. (Lower rows in the matrix represent higher delegation strengths). These elements, $\mu_{Q}\left(x\right),$ are functions of x: $\mu_{Q}\left(x\right)=a\cdot x+b\cdot x^{2}+c$, representing the entries of the Hamiltonian matrix that push the decision-maker’s delegation strength to the higher level of delegation strength (lower rows). In this study, for $\mu_{Q}\left(x\right)$, a simple linear function (b=0 and c=0) is used, where $\mu_{Q}\left(x\right)=a\cdot x$.

The off-diagonal elements, such as the diffusion rate, $\sigma_{Q}$, control the diffusion amplitudes that capture the dynamics of flowing out from the basis state; the diffusion rate induces uncertainty over different delegation strength levels. In the context of trust, trust in AI will push the decision-maker’s delegation state to the lower rows, whereas distrust will try to pull it back to the upper rows.
(51)$$H=\left[\begin{array}{cc} \begin{array}{ccc} \mu_{Q}\left(1\right) & \sigma_{Q} & \;\;\;\;\;\;0\\ \sigma_{Q} & \mu_{Q}\left(2\right) & {\;\;\;\;\;\;\sigma }_{Q}\\ 0 & \sigma_{Q} & \;\;\;\;\;\;\ddots \end{array}\;\; & \;\;\;\;\;\;\;\;\;\;\;\;\;\;\begin{array}{ccc} \cdots \;\;\;\;\;\;\;\;\;\; & \;\;\;\;\;\;\;\;\;\;\;\;\;\;\;0 & \;\;\;\;\;\;\;\;\;\;\;\;\;\;0\\ 0\;\;\;\;\;\;\;\;\; & \;\;\;\;\;\;\;\;\;\;\;\;\;\;\;\vdots & \;\;\;\;\;\;\;\;\;\;\;\;\;\;0\\ \ddots \;\;\;\;\;\;\;\;\; & \;\;\;\;\;\;\;\;\;\;\;\;\;\;\;\;0 & \;\;\;\;\;\;\;\;\;\;\;\;\;\vdots \end{array}\\ \begin{array}{ccc} \vdots & \;\;\;\;\;\;\;\;\;0 & \;\;\;\;\;\;\;\;\ddots \\ 0 & \;\;\;\;\;\;\;\;\;\;\vdots & \;\;\;\;\;\;\;\;\;0\\ 0 & \;\;\;\;\;\;\;\;\;\;\;0\; & \;\;\;\;\;\;\;\;\;\cdots \end{array} & \;\;\;\;\;\;\;\;\;\;\;\;\begin{array}{ccc} \mu_{Q}\left(x-2\right)\;\;\;\;\;\;\;\;\; & \sigma_{Q} & 0\\ \sigma_{Q}\;\;\;\;\;\;\;\;\;\;\;\; & \mu_{Q}\left(x-1\right)\; & \sigma_{Q}\\ 0\;\;\;\;\;\;\;\;\;\;\;\; & \;\;\sigma_{Q} & \mu_{Q}\left(X\right) \end{array} \end{array}\;\;\right]$$
(52)$$K=\left[\begin{array}{cc} \begin{array}{ccc} -\sigma_{M}-\mu_{M} & \sigma_{M}-\mu_{M} & 0\\ \sigma_{M}+\mu_{M} & -2\sigma_{M} & \sigma_{M}-\mu_{M}\\ 0 & \sigma_{M}+\mu_{M} & \ddots \end{array}\;\; & \;\;\;\;\;\begin{array}{ccc} \cdots \;\;\;\;\;\;\; & \;\;\;\;\;\;\;\;\;\;\;\;\;\;\;\;\;\;0 & \;\;\;\;\;\;\;\;\;\;\;\;\;\;\;\;\;\;\;\;0\\ 0\;\;\;\;\;\;\; & \;\;\;\;\;\;\;\;\;\;\;\;\;\;\;\;\;\;\vdots & \;\;\;\;\;\;\;\;\;\;\;\;\;\;\;\;\;\;\;\;0\\ \ddots \;\;\;\;\;\;\;\; & \;\;\;\;\;\;\;\;\;\;\;\;\;\;\;\;\;\;0 & \;\;\;\;\;\;\;\;\;\;\;\;\;\;\;\;\;\;\;\;\vdots \end{array}\\ \begin{array}{ccc} \vdots & \;\;\;\;\;\;\;\;\;\;\;\;\;\;0 & \;\;\;\;\;\;\;\;\;\;\;\;\;\;\ddots \\ 0 & \;\;\;\;\;\;\;\;\;\;\;\;\;\;\vdots & \;\;\;\;\;\;\;\;\;\;\;\;\;\;0\\ 0 & \;\;\;\;\;\;\;\;\;\;\;\;\;\;\;0\; & \;\;\;\;\;\;\;\;\;\;\;\;\;\;\cdots \end{array} & \;\;\;\;\;\;\;\;\;\;\begin{array}{ccc} -2\sigma_{M}\;\;\;\;\;\;\;\;\; & \sigma_{M}-\mu_{M} & 0\\ \sigma_{M}-\mu_{M}\;\;\;\;\;\;\;\;\;\;\;\; & -2\sigma_{M} & \sigma_{M}+\mu_{M}\\ 0\;\;\;\;\;\;\;\;\;\;\;\; & \;\;\sigma_{M}+\mu_{M} & -\sigma_{M}-\mu_{M} \end{array} \end{array}\;\;\right]$$

The drift rate, $\mu_{M}$, represents the dynamics of the intensity matrix, *K* (52), which pushes the delegation strength to the higher strength level; in this case, lower rows in the matrix represent higher strength levels. The diffusion rate is similar to the Hamiltonian; this introduces dispersion to the delegation strength, which captures some of the inconsistencies in decision-making. The difference between the Hamiltonian (*H*) and the intensity matrix (*K*) is that the diffusion rate in the intensity matrix is dispersed via the probability distribution, whereas in the *H* case, it is conducted via amplitudes.

The Hilbert space dimension choice for the open system model is a critical decision in building the model because the dimension can become computationally expensive. For example, running a 101-dimensional open system model is computationally cumbersome. Following the discussions in [14,59], we reduced the number of dimensions and ran various analyses using models that ranged between 9 and 31 dimensions. Similar to [14,59], a 21-dimensional model was chosen for the presented results in this paper. Since the fitted values are mean delegation states, choosing higher or lower dimensional models did not affect the results.

The main challenge in building an open system model of a cognitive phenomenon is choosing the Γ matrix’s coefficients, $\gamma_{ij}$. Depending on the choice, the Γ matrix is constrained by different requirements. For example, following the discussion in [80], if we choose the $\gamma_{ij}$ to be the transition probabilities in the Markov transition matrix, $T_{ij}\left(t=\varepsilon \right)$, a direct connection to Markov dynamics can be achieved. Then, by choosing α=0, Equation (40) reduces to Equation (41). After following the vectorized solutions provided by [61], as well as, [85], the solution to Equation (41) provides the following:(53)$$\frac{d\rho_{kk}\left(t\right)}{dt}=\sum_{j}\rho_{jj}\cdot \gamma_{kj}-\sum_{i}\gamma_{ik}\cdot \rho_{kk}$$

Setting up $\gamma_{ij}=T_{ij}\left(\varepsilon \right)$ requires that the entries within each column in $\gamma_{ij}$ add up to one because $T\left(t\right)$ is restricted by single stochasticity. Following the discussion in [80], since a Markov process is based on the intensity matrix (*K*), which obeys the Kolmogorov–Chapman solution, $\frac{d\phi (t)}{dt}=K\cdot \phi \left(t\right)$, the solution to Equation (53), which is $\frac{d\phi (t)}{dt}=(T\left(\varepsilon \right)-I)\cdot \phi \left(t\right)$, results in incompatibility. Therefore, it may not fully capture the Markov dynamics.

As discussed by [61], setting the $\gamma_{ij}$ to the intensity matrix can work as a solution. However, this introduces another challenge that requires the columns in $\gamma_{ij}$ to sum to zero because an intensity matrix is a negative definite matrix. As a result, a density matrix cannot be maintained for the entire time interval. The best solution provided in [80] is to set $\gamma_{ij}=\left(\frac{T\left(\varepsilon \right)}{\varepsilon }\right)$, which provides a solution to the issues that arise in the previous two solutions. However, for very small values of ε, the interference terms and off-diagonal terms rapidly dissipate.

### 5.2. Exploratory Analysis

The experimental results provide the groundwork for tuning the parameters of the quantum open system equation. These results demonstrate that setting the $\gamma_{ij}$ component of the Lindblad operator equal to the intensity matrix provides the best results as seen in Figure 11. With these parameters, it is now possible to model both choice and no-choice conditions with one equation. The fit parameters used for the Hamiltonian matrix, intensity matrix, and alpha parameter are provided in Table 4. However, capturing the earlier timings proved difficult because of the sheer number of combinations that would have to be attempted to tune the equation further. However, the sum of squared errors (SSE) was improved by 54.8% (a difference of 226 in previous modeling results). The SSE is provided in Table 5. Future work is set to develop machine learning models that can tune the parameters more efficiently. Nevertheless, a quantum open system modeling approach shows promise for modeling human behavior when interacting with AI.

The interplay between Markov and quantum dynamics provides a potentially new approach for modeling decision-making dynamics in general and decision-making with AI in particular. If human decision-making does indeed follow a more quantum open systems approach, developing more optimal machine policies for interacting with humans may yield more interesting results [70]. While the results of this study appear promising, more work is needed to flush out more details and to test the bounds of how far these techniques may generalize.

## 6. Discussion

Different considerations are needed for how human and AI decision-making are conceptualized within HITL constructs. If AI takes over additional awareness/decision-making spaces, as projected in the literature and media, researchers will need to carefully consider how humans and AI are integrated to improve human decision-making in light of these advancements.

Significant work lies ahead for developing HITL-AI systems. Incorporating humans into AI-augmented situational awareness and decision-making (or vice versa) will take many different forms. It is clear from the research that humans and AI systems will continue to engage in shared decision-making [87,88]; the questions will be what decisions are ceded to AI and how will organizations align their different rationalities. However, capitalizing on QPT-based research findings in the design of HITL-AI systems opens the door for reevaluating previous research. For instance, earlier research, such as the belief-adjustment model [89], which found order effects due to recency bias or weighting information based on temporal arrival, could be reevaluated with QPT. Without capturing the correct dynamics, HITL-AI systems will exacerbate decision cycles as engineers attempt to reconcile human and AI rationalities. Future research will need to address how HITL-AI systems operate as time pressures increase and what can be done to improve decision-making in high-tempo and ethically significant operations with comprehensive frameworks.

Design considerations that capitalize on QPT-based decision models to improve human and machine interactions are still in their infancy. Some researchers have suggested that QPT formalisms can be applied by machines to help cognitively impaired humans (e.g., dementia or Alzheimer’s) achieve specific goals (e.g., washing hands, taking medications) [70]. Yet, the same considerations can also apply to HITL-AI systems. Knowing the shortcomings of human reasoning and information processing, machines can better model human cognitive processes with QPT to account for how (1) humans are influenced by the order of information (e.g., humans should decide before AI reveals its own perspective, knowing when and whether to solicit for AI advice would lead to different decision outcomes [90]; (2) new pieces of information can result in incompatible perspectives and higher ontic uncertainty between agents. Considerations for these design parameters could improve engineering AI systems for decision-making through better predictability of human–machine interactions. Consequently, HITL-AI systems may be engineered to move human decision-making toward a more Bayesian optimal choice [70]. Quantum open systems have been shown to be potential pathways for modeling human–AI interactions in decision-making. With these design considerations in mind, much work still lies ahead.

### 6.1. Limitations

This study has a few limitations, which may bind generalizability and applicability to similar domains of interest. First, the imagery analysis task did not require any decisions of consequence. This artificiality could lead to a shortfall in generalizing to real-world employment scenarios, which entails a significant risk calculus or additional decision considerations. Secondly, the system used for annotating images was a notional AI system that was only tested for 21 trials. However, many AI systems today operate in real-time and can annotate live video. For this reason, participant behaviors may differ with the additional context supplied by live video. Therefore, we stress caution when generalizing beyond static source materials in this decision-making experiment supported by AI.

### 6.2. Future Research

The phenomenon of understanding trust in decision-making with AI is still a nascent area given the ever-increasing capability of machines. A recent study [91] suggests that new research is needed to “examine trust-enabled outcomes that emerge from dyadic team interactions and extend that work into how trust evolves in larger teams and multi-echelon networks” (p. 3). While this study takes a quantitative approach to further elucidate choice and timing as variables that influence the level of trust, additional research is needed. For instance, do the effects of intermediate judgments dissipate after a period of time? Would choice decisions be different if the incentive/reward structure was set up differently (i.e., loss of compensation for incorrect decisions)? While this experiment provides a good first approximation, it is far from being an endpoint in this area of research. Moreover, continued research along these lines will hopefully yield additional findings that can help explicate trust in AI-supported decision-making.

## 7. Summary

The use of the quantum probability theory to augment models of human–AI decision-making holds much promise. Mathematical formalisms of quantum theory are, in fact, the most accurate theories ever tested [92]. QPT and similar efforts to formulate a concept of quantum decision theory (QDT) have provided novel results that can better model uncertainty and human decision-making behaviors. Applying QPT to human and machine situational awareness models is still in the nascent stage of development for the human–machine dyad [93,94,95]. QPT modeling can ameliorate interactions by providing a novel way to capture diverse types of uncertainty within human–AI decision systems and, therefore, improve human–machine engineering efforts. For these reasons, quantum open systems hold much promise for better modeling human–AI decision-making like never before.

## Figures and Tables

**Figure 1 entropy-26-00500-f001:**
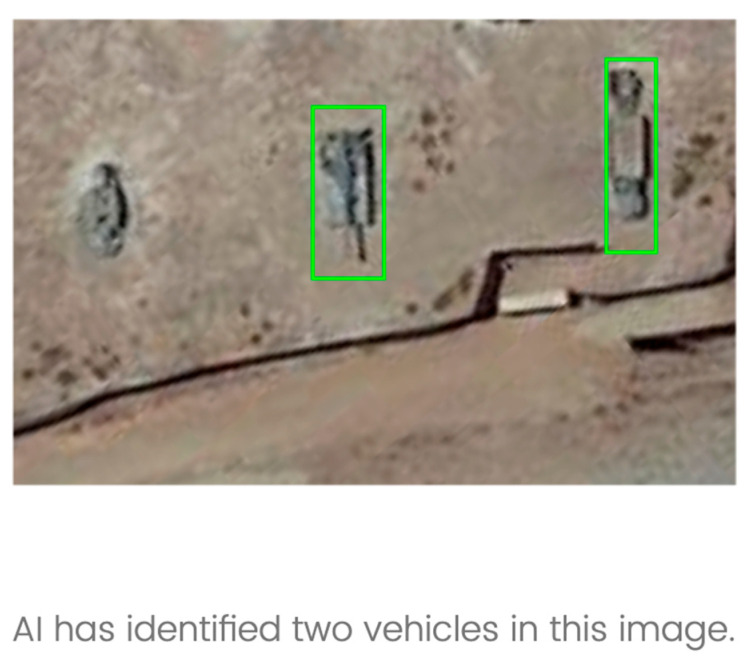
Example AI-annotated imagery from the study.

**Figure 2 entropy-26-00500-f002:**
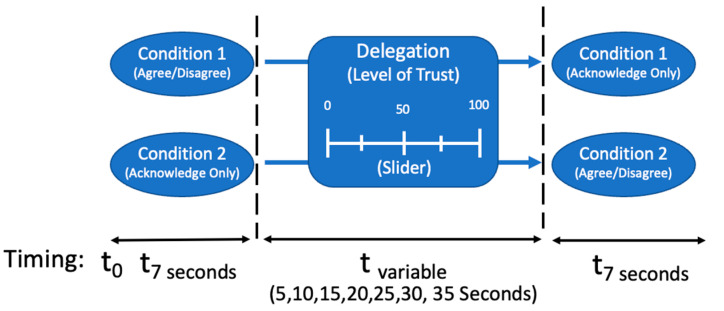
Experimental design. From Ref. [60].

**Figure 3 entropy-26-00500-f003:**
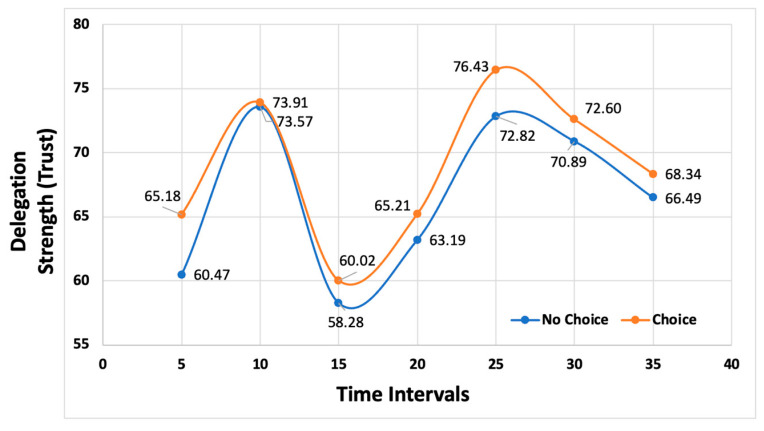
Comparison of mean delegation strength by condition and timing.

**Figure 4 entropy-26-00500-f004:**
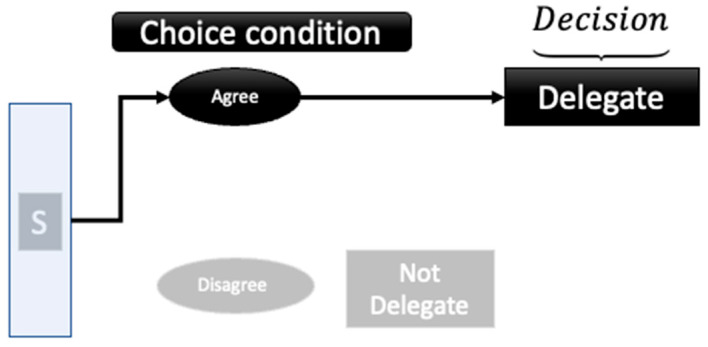
Choice condition ‘agree’—delegate models.

**Figure 5 entropy-26-00500-f005:**
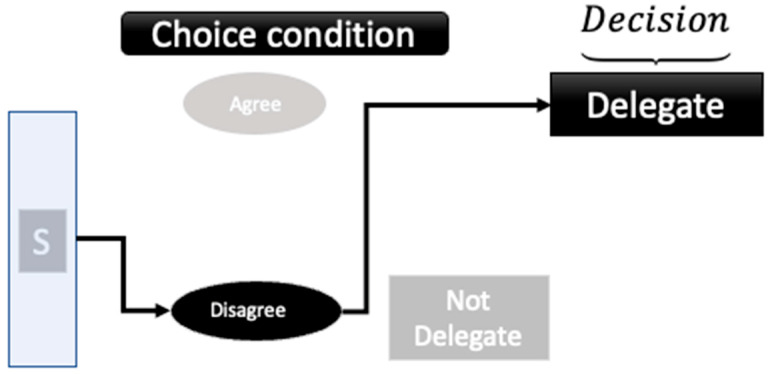
Choice condition ‘disagree’—delegate.

**Figure 6 entropy-26-00500-f006:**
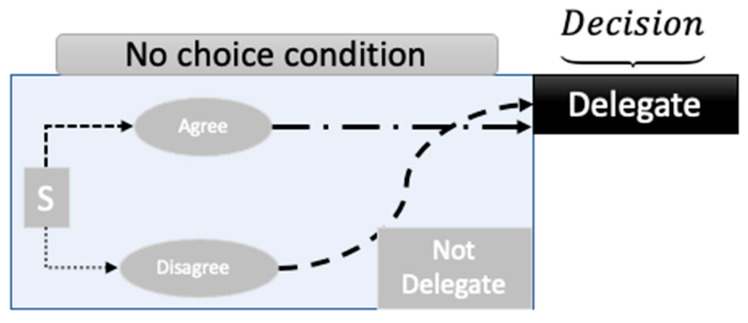
No-choice condition decision = delegate.

**Figure 7 entropy-26-00500-f007:**
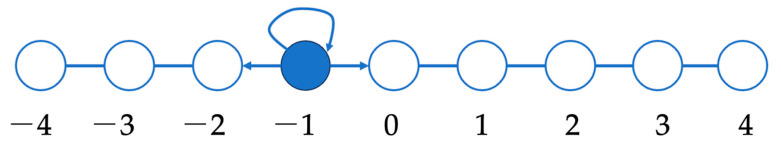
State transition for a random walk model that captures a 9-state transition. Adapted from [58,61].

**Figure 8 entropy-26-00500-f008:**
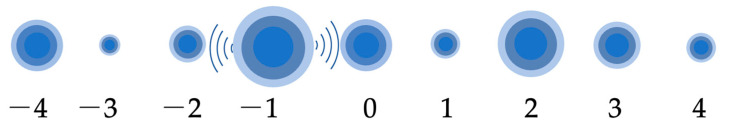
Quantum superposition of all nine states. Adapted from [58,61].

**Figure 9 entropy-26-00500-f009:**
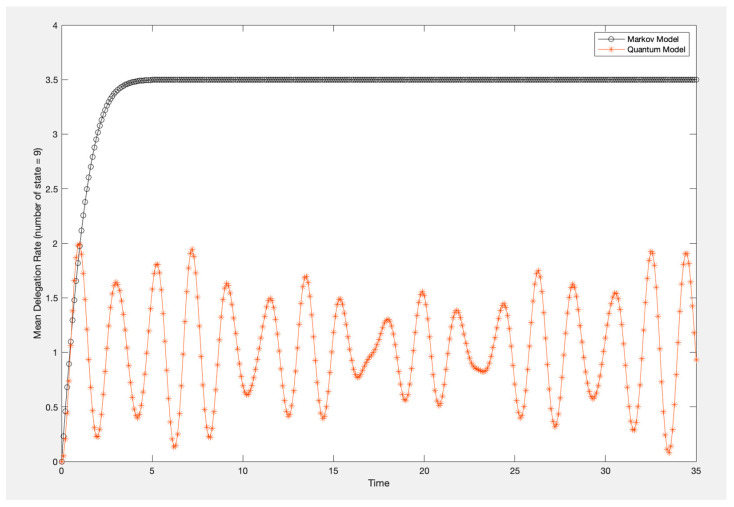
Nine-state quantum and Markov models.

**Figure 10 entropy-26-00500-f010:**
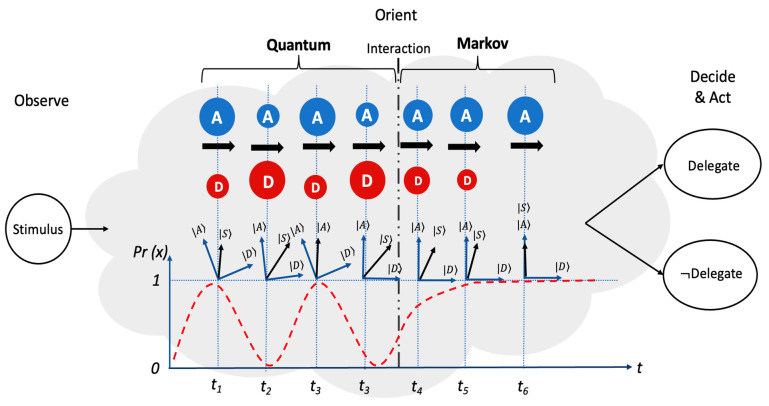
Quantum open systems modeling approach.

**Figure 11 entropy-26-00500-f011:**
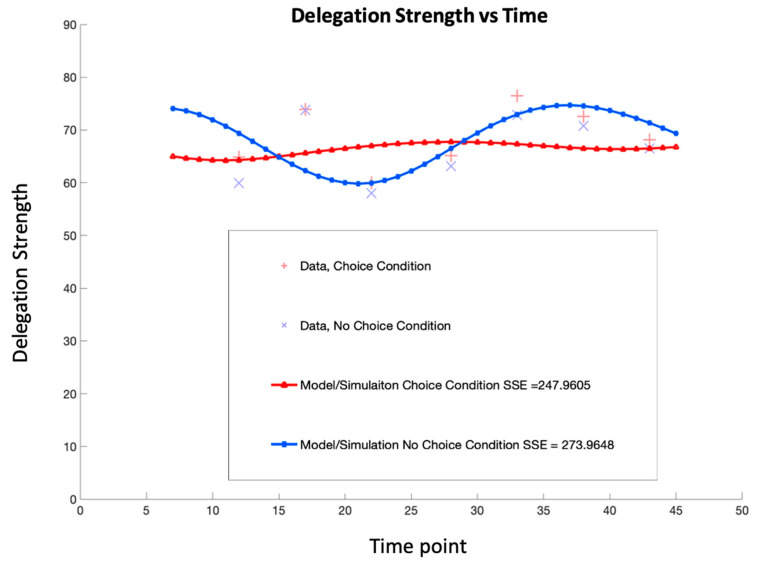
Quantum open system modeling of delegation strength vs. time results for choice and no-choice conditions.

**Table 1 entropy-26-00500-t001:** Experimental results.

Categorize, Then Delegate Conditions							Delegate Only
Timing	*Pr*(A)	*Pr*(Del|A)	*Pr*(Dis)	*Pr*(Del|Dis)	TP (Del) Intermediate Judgment	*Pr*(Del) No Intermediate Judgment	*Pr*(Del)
5 s	0.7097	0.5966	0.2903	0.1389	0.4637	0.4118	−0.0519
10 s	0.8495	0.7089	0.1505	0.2143	0.6344	0.6097	−0.0247
15 s	0.6231	0.6173	0.3769	0.2143	0.4654	0.4559	−0.0095
20 s	0.6833	0.6042	0.3167	0.2360	0.4875	0.4926	0.0050
25 s	0.8566	0.7225	0.1434	0.2895	0.6604	0.6182	−0.0422
30 s	0.8327	0.7143	0.1673	0.1556	0.6208	0.5941	−0.0267
35 s	0.7907	0.6716	0.2093	0.1296	0.5581	0.5257	−0.0324

Table legend—A: agree; Del: delegate; Dis: disagree; TP: total probability.

**Table 2 entropy-26-00500-t002:** Joint probability table for delegate, not-delegate, agree, and disagree events.

	Agree (*A*)	Disagree (*DisA*)
Delegate (*D*)	*a*	*b*
Not Delegate (*notD*)	*c*	*d*

**Table 3 entropy-26-00500-t003:** Probability of a table for ‘delegate’ or ‘not delegate’ without including choices for ‘agree’ and ‘disagree’.

Delegate (*D*), Pr(D)	e
Not Delegate (*notD*), $\mathrm{Pr}\left(notD\right)$	f

**Table 4 entropy-26-00500-t004:** Fit parameters for the quantum open system equation.

Fit Parameters	
$\mu_{Q}$	390.45
$\sigma_{Q}$	30.12
$\mu_{M}$	5.95
$\sigma_{M}$	19.62
α	0.21

**Table 5 entropy-26-00500-t005:** Quantum open system modeling the results for SSE.

Condition	SSE for Quantum Open System Models
Choice	247.9605
No Choice	273.9648

## Data Availability

The data presented in this study are available on request from the corresponding author. Restrictions may apply.

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
