# Peer review of "Intermediate Judgments and Trust in Artificial Intelligence-Supported Decision-Making"

_entropy, 2024, doi:10.3390/e26060500_

Round 1

Reviewer 1 Report

Comments and Suggestions for Authors

This article presents an interesting and novel approach to modelling trust in human-AI interactions. 

A notable aspect of the article is the use of open quantum systems (OQS), which are the cutting edge of modeling the dynamics of decision making using a quantum approach.

The experimental design is thoughtfully constructed. In particular, the agree/disagree condition is an interesting element that is both conceptually appropriate for the examination of trust as well as allowing empirical testing whether the law of total probability is violated. 

My major issue is reconciling how the agree/disagree dynamics (n=2) relate the quantum random walk models (n=9), which are used as the quantum part of the OQS. The exploratory analysis (Section 5.2) lacks sufficient detail to judge its worth.

The paper also has potential tutorial value. If the issues detailed below are clarified, then others can benefit from this paper by seeing how to define core elements prescribing the dynamics of OQS models, namely the Hamiltonian H and the intensities used in the Linblad operators, based on a particular experimental setting.

Minor comments are inserted at annotations in the attached pdf 

Comments on the Quality of English Language

A couple of minor issues which have been highlighted in the attached pdf

Author Response

Thank you very much for the thorough feedback. A detailed comments matrix is provided for your review.

Reviewer 2 Report

Comments and Suggestions for Authors

This is a well-written, clear paper that very competently propose to use advanced tools of quantum decision theory to consider the interaction between humans and AI in decision-making.  I would suggest to make reference to Quantum persuasion  (Danilov and Lambert Mogiliansky, (2018 JME) and well as Danilov et al. 2018 (Decision Theory)) which are dealing with the use of information with quantum like agents.  

While the experiment has an interesting design, the use of MTurk involving a biased sample is problematic and the results on the impact of intermediate judgement are not convincing ( negligible interference effects). 

Author Response

Thank you for the feedback.  A detailed comments matrix is provided for your review.
